# Pathologic and molecular responses to neoadjuvant trastuzumab and/or lapatinib from a phase II randomized trial in HER2-positive breast cancer (TRIO-US B07)

Sara A. Hurvitz et al.[#]

In this multicenter, open-label, randomized phase II investigator-sponsored neoadjuvant trial with funding provided by Sanofi and GlaxoSmithKline (TRIO-US B07, Clinical Trials NCT00769470), participants with early-stage HER2-positive breast cancer ($N = 128$) were recruited from 13 United States oncology centers throughout the Translational Research in Oncology network. Participants were randomized to receive trastuzumab (T; $N = 34$), lapatinib (L; $N = 36$), or both (TL; $N = 58$) as HER2-targeted therapy, with each participant given one cycle of this designated anti-HER2 therapy alone followed by six cycles of standard combination chemotherapy with the same anti-HER2 therapy. The primary objective was to estimate the rate of pathologic complete response (pCR) at the time of surgery in each of the three arms. In the intent-to-treat population, we observed similar pCR rates between T (47%, 95% confidence interval [CI] 30–65%) and TL (52%, 95% CI 38–65%), and a lower pCR rate with L (25%, 95% CI 13–43%). In the T arm, 100% of participants completed all protocol-specified treatment prior to surgery, as compared to 69% in the L arm and 74% in the TL arm. Tumor or tumor bed tissue was collected whenever possible pre-treatment ($N = 110$), after one cycle of HER2-targeted therapy alone ($N = 89$), and at time of surgery ($N = 59$). Higher-level amplification of HER2 and hormone receptor (HR)-negative status were associated with a higher pCR rate. Large shifts in the tumor, immune, and stromal gene expression occurred after one cycle of HER2-targeted therapy. In contrast to pCR rates, the L-containing arms exhibited greater proliferation reduction than T at this timepoint. Immune expression signatures increased in all arms after one cycle of HER2-targeted therapy, decreasing again by the time of surgery. Our results inform approaches to early assessment of sensitivity to anti-HER2 therapy and shed light on the role of the immune microenvironment in response to HER2-targeted agents.

[#]A list of authors and their affiliations appears at the end of the paper.

Although trastuzumab substantially improves disease-free survival for patients with HER2-positive (HER2+) breast cancer[1–5], approximately one-quarter of trastuzumab-treated patients with early-stage disease experience recurrence during the first decade, signifying that treatment resistance remains a challenge[6,7]. The achievement of a pathologic complete response (pCR) after neoadjuvant therapy appears to be a surrogate marker for disease-related outcomes including event-free survival (EFS) or overall survival (OS) in HER2+ or triple-negative subtypes[8,9]. Thus, the neoadjuvant setting is increasingly utilized for the clinical investigation of promising new therapies[10]. When added to neoadjuvant chemotherapy, trastuzumab has been shown to improve pCR rates[11–14] and EFS[12]. Synergistic interactions between trastuzumab and the oral HER1/HER2-targeted tyrosine kinase inhibitor lapatinib have been observed in HER2-overexpressing cell lines[15,16] and in HER2+ tumor xenograft models[17]. Clinically, dual HER2 inhibition using lapatinib and trastuzumab led to significant improvement in OS compared to lapatinib alone in the metastatic setting[18,19]. Several neoadjuvant trials have compared regimens containing dual HER2-blockade utilizing trastuzumab and lapatinib to regimens with only trastuzumab or lapatinib[20–25]. While all studies showed numerically higher pCR rates using dual HER2-blockade compared to trastuzumab, only two[20,22] reached a statistically significant difference. These trials differed based on chemotherapy backbone, treatment order, duration of HER2-directed therapy, and definition of pCR, which could partially explain these conflicting results.

A major goal of the translational science from these and other trials of HER2-targeted therapy has been to identify tumors that are highly sensitive to treatment, such that therapy can be potentially de-escalated, as well as those with primary resistance, where new strategies may be needed. Multiple putative biomarkers of HER2-targeted therapy response have emerged from these efforts, but there is no validated biomarker that predicts pCR with adequate accuracy to allow patient stratification. Lower estrogen receptor (ER) expression[26,27] and higher HER2 amplification or expression[26–28] are associated with a higher pCR rate. Using PAM50 expression-based intrinsic subtyping, tumors that classify as the HER2-enriched intrinsic subtype display a higher rate of pCR after regimens containing HER2-targeted therapy than HER2+ tumors that classify as other intrinsic subtypes[26,27,29–31]. Tumors with evidence of immune activation (tumor-infiltrating lymphocytes [TILs][29,32,33] or higher expression of immune gene signatures[26,27,29,31,33]) also have higher rates of pCR. These varying response signatures interrelate with each other, and while using multiple signatures together increases the ability to predict pCR[34], how each may independently contribute to HER2-targeted therapy sensitivity or resistance is uncertain. It is also unknown how the tumor may change across HER2-targeted therapies: one study of the combination of trastuzumab and lapatinib suggested that a change in intrinsic subtype to normal-like after a short course of HER2-targeted therapy might predict pCR[30] and that, with the combination of HER2-targeted therapy and endocrine therapy, HR+ /HER2+ tumors frequently converted to the luminal A subtype[35]. However, to date, very few studies have assessed change in tumor cell gene expression and microenvironmental composition throughout a period of HER2-targeted therapy.

Here we report the results of phase II randomized neoadjuvant trial aimed at evaluating the pCR rates associated with trastuzumab and/or lapatinib in combination with chemotherapy. In this study, tumor (or tumor bed) samples were collected prior to treatment, after one cycle of run-in HER2-targeted therapy (without chemotherapy), and at surgery after completion of HER2-targeted therapy with chemotherapy added. The histopathologic and expression data from these samples allow assessment of how biomarkers correlate with one another and validation of how they perform as predictors of pCR. Importantly, the on-treatment biopsies were also used to assess how these biomarkers, signaling pathways, and microenvironmental composition change throughout HER2-targeted therapy.

## Results

**Comparative pathologic complete response (pCR) rates.** Women ages 18–70 with anatomic stage I–III unilateral HER2+ breast carcinoma were eligible for Translational Research in Oncology (TRIO)-B07, registered as NCT00769470 (www.clinicaltrials.gov). The trial enrolled participants at 13 centers in the United States, with the first participant enrolled 12/22/2008 and the final enrolled 12/20/2012; enrollment ended after four years with 130 of 140 planned participants. The first 20 participants were allocated to trastuzumab and lapatinib (TL) and the next 110 participants were randomized to one of three arms, each with one run-in cycle of HER2-targeted therapy alone followed by six cycles of docetaxel and carboplatin combined with the same HER2-targeted therapy to which they were randomized: trastuzumab (T), lapatinib (L), or trastuzumab and lapatinib (TL) (Fig. 1). Two participants withdrew from the study prior to starting any treatment and were excluded from efficacy and safety analyses. Of 128 participants, 25 came off their assigned study treatment prior to surgery, but still completed surgery (10 in L, 15 in TL). One participant (L) did not complete surgery. All 128 participants were included in the intent to treat (ITT) analyses. Baseline characteristics (Table 1) were well balanced between the three treatment arms. The median patient age was 48 (range 27–78). Hormone receptors were negative (HR−) in 56 participants (44%) and were ER and/or progesterone receptor-positive (HR+) in 72 (56%). At presentation, 6 (5%) participants had clinical anatomic stage I, 86 (67%) stage II, and 36 (28%) stage III breast cancer.

Pathologic complete response (pCR), defined as an absence of viable invasive tumor cells in the breast and examined axillary lymph nodes at the time of definitive surgery, was the primary endpoint of the study. The overall pCR rate for the ITT population was 43% (95% confidence interval (CI) 34–52%), including 47% (95% CI 30–65%) in T, 25% (95% CI 13–43%) in L, and 52% (95% CI 38–65%) in TL (Table 2). Using pairwise comparisons ($\chi 2$ tests), pCR was significantly lower in L than in TL ($p = 0.01$) but no statistically significant differences were detected between T and L ($p = 0.14$) or T and TL ($p = 0.88$). In exploratory analyses by hormone receptor status, HR + tumors had a lower pCR rate than HR- (33% (95% CI 23–46%) vs. 55% (95% CI 42–68%), respectively). No significant differences in pCR were observed between arms in the HR− subset; however, in the HR+ subset, L had a significantly lower pCR rate compared to T ($p = 0.04$) and TL ($p = 0.03$).

**Safety and tolerability.** Overall, 80% of participants completed all protocol-specified treatment prior to surgery: 100% (34/34) in T, 69% (25/36) in L, and 74% (43/58) in Arm TL. In lapatinib-containing arms, 22% of participants stopped neoadjuvant therapy due to adverse events (8/36 in L, 13/58 in TL). An additional five patients came off study prior to completing protocol-specified therapy due to consent withdrawal ($N = 2$ L, $N = 1$ TL), progressive disease ($N = 1$, TL), or non-adherence ($N = 1$, L). The relative dose intensity (averaged for all treatment combined in each arm) was 98% in Arm T, 85% in Arm L, and 86% in Arm TL.

There were no deaths and no episodes of congestive heart failure reported. Left ventricular ejection fraction decreased >10%

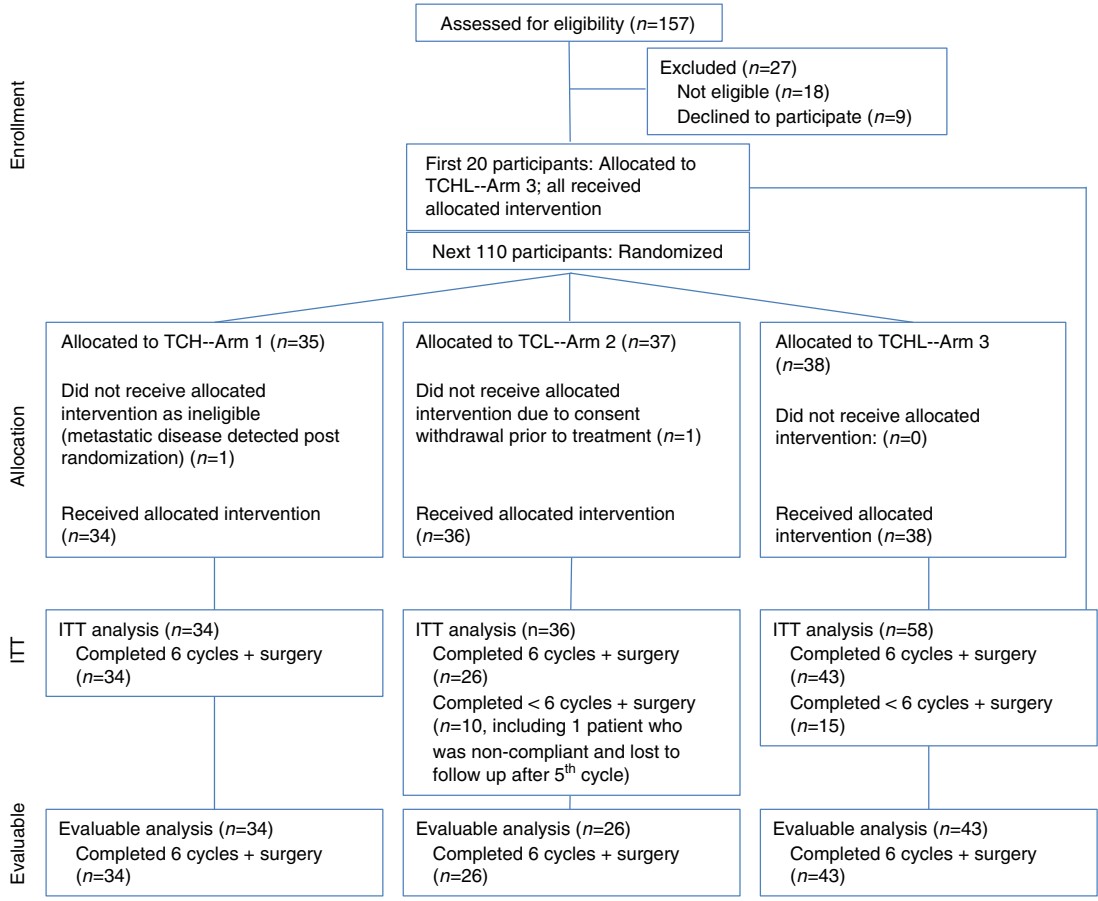

**Fig. 1 TRIO-US B07 clinical trial participants.** 130 participants were enrolled across three treatment arms. All participants received docetaxel plus carboplatin (TC) every 3 weeks. In addition, participants in Arm 1 received trastuzumab (TCH), Arm 2 received lapatinib (TCL), and Arm 3 received both trastuzumab and lapatinib (TCHL). Two participants withdrew from the study prior to starting any treatment, leaving 128 participants remaining in the intent to treat (ITT) population. Of 128 participants, 25 came off study treatment prior to surgery (10 in Arm 2, 15 Arm 3), leaving 103 participants included in the evaluable analysis.

| Table 1 Baseline characteristics. | | | |
|---|---|---|---|
| **Participant characteristics** | **Treatment arm** | | |
| | **Arm 1: TCH** (n = 34) n (%) | **Arm 2: TCL** (n = 36) n (%) | **Arm 3: TCHL** (n = 58) n (%) |
| Median age | 48 | 51 | 47 |
| Histology | | | |
| Invasive Ductal | 33 (97) | 34 (94) | 52 (90) |
| Invasive Lobular | 1 (3) | 0 (0) | 4 (7) |
| Other | 0 (0) | 2* (6) | 2** (3) |
| Hormone Receptor Status | | | |
| ER− and PR− | 14 (41) | 18 (50) | 24 (41) |
| ER+ and/or PR+ | 20 (59) | 18 (50) | 34 (59) |
| Median Tumor Size (cm) | 5.54 (1–20) | 5.16 (2–14) | 4.15 (1.4–12) |
| Tumor Size | | | |
| ≤3 cm | 11 (32) | 11 (31) | 17 (29) |
| >3 cm | 23 (68) | 25 (69) | 41 (71) |
| Stage | | | |
| I | 2 (6) | 1 (3) | 3 (5) |
| II | 20 (59) | 28 (78) | 38 (66) |
| III | 12 (35) | 7 (19) | 17 (29) |

ER: estrogen receptor; PR: progesterone receptor; T: docetaxel; C: carboplatin; H: trastuzumab; L: lapatinib.
* One multifocal invasive carcinoma and one mammary not otherwise specified.
** One invasive micropapillary carcinoma and one invasive undifferentiated carcinoma.

from baseline and below the lower limits of normal in 5 patients (1 patient in T, 2 in L, 1 in TL). The most common grade ≥3 AEs in Arms T/L/TL respectively were pain (9%, 19%, 19%), diarrhea (3%, 14%, 28%), neutropenia (12%, 14%, 14%), anemia (9%, 8%, 7%), hypokalemia (6%, 6%, 9%), and infection (6%, 14%, 9%) (Table 2).

**Pre-treatment tumor clinical variables and subtypes.** All molecular analyses were performed in the expression-evaluable cohort, which omitted samples with no biopsy or insufficient RNA quantity or quality and consisted of 110 participants with a pre-treatment tumor biopsy, 89 participants with an on-treatment biopsy after one cycle of HER2-targeted therapy (matched to pre-treatment biopsy), and 59 participants with a post-treatment excision specimen (matched to pre-treatment biopsy). Distribution of treatment arms, HR status, and pCR outcomes were similar between the expression-evaluable cohorts and the overall cohort (Supplementary Table 1).

We began by examining the associations between pre-treatment clinical and expression-based subtypes and pCR (Fig. 2a, Supplementary Table 2). HR-status was associated with a higher pCR rate (odds ratio (OR) 2.3 (95% CI 1.0–5.0)), as was strong (3+) HER2 immunohistochemical (IHC) staining (OR 11.9 (95% CI 2.1–304)) and higher HER2 fluorescent in situ hybridization (FISH) ratio (the ratio of HER2 copy number to centromere 17 copy number) ($\beta = 0.13$ (95% CI 0.01–0.26)). The association between HER2 FISH ratio and pCR was driven by a

**Table 2 Pathologic complete response rates and toxicities.**

|  | Arm 1: TCH N % (95% CI) | Arm 2: TCL N % (95% CI) | Arm 3: TCHL N % (95% CI) | P value (Chi squared) |
|---|---|---|---|---|
| pCR (pT0/ispN0) ITT | 16/34<br>47 (30–65) | 9/36<br>25 (13–43) | 30/58<br>52 (38–65) | **Arm 2 vs 3: 0.01**<br>Arm 1 vs 2: 0.14<br>Arm 1 vs 3: 0.88 |
| pCR by HR status |  |  |  |  |
| ER− and PR− (N = 56) | 8/14<br>57 (30–82) | 7/18<br>39 (18–64) | 16/24<br>67 (45–84) | Arm 2 vs 3: 0.07<br>Arm 1 vs 2: 0.30<br>Arm 1 vs 3: 0.55 |
| ER+ and/or PR+ (N = 72) | 8/20<br>40 (20–64) | 2/18<br>11 (2–36) | 14/34<br>41 (25–59) | **Arm 2 vs 3: 0.03**<br>**Arm 1 vs. 2: 0.04**<br>Arm 1 vs 3: 0.93 |
| pCR evaluable participants | 16/34<br>47 (30–65) | 7/25<br>28 (13–50) | 21/43<br>49 (34–64) | Arm 2 vs 3: 0.09<br>Arm 1 vs 2: 0.14<br>Arm 1 vs 3: 0.88 |
| Completion rates | 34/34<br>100 (87–100) | 25/36<br>69 (52–83) | 43/58<br>74 (61–84) |  |
| Grade ≥ 3 Toxicity ≥ 5% | N (%) | N (%) | N (%) |  |
| Diarrhea* | 1 (3) | 5 (14) | 16 (28) | Arm 1 vs 3: 0.01 |
| Pain | 3 (9) | 7 (19) | 11 (19) |  |
| Neutropenia | 4 (12) | 5 (14) | 8 (14) |  |
| Infection | 2 (6) | 5 (14) | 5 (9) |  |
| Anemia | 3 (9) | 3 (8) | 4 (7) |  |
| Hypokalemia | 2 (6) | 2 (6) | 5 (9) |  |
| Fatigue | 2 (6) | 3 (8) | 3 (5) |  |
| Dehydration | 1 (3) | 0 (0) | 5 (9) |  |
| Thrombocytopenia | 1 (3) | 3 (8) | 2 (3) |  |

ER: estrogen receptor; HR: hormone receptor; ITT: intent-to-treat; pCR: pathologic complete response (breast and lymph nodes); PR: progesterone receptor; pts: participants; T: docetaxel; C: carboplatin; H: trastuzumab; L: lapatinib; CI: confidence interval.
P-values < 0.05 are bolded. * Significantly different.

high pCR rate in tumors with very high HER2 FISH ratios: the pCR rate among the 8.6% of tumors with FISH ratio ≥ 12 was 85.7%, while among tumors with FISH ratio < 12 it was 39.2% with no association between FISH ratio and pCR in that group ($p = 0.41$).

Despite HER2 amplification based on FISH as determined locally per ASCO-CAP 2007 guidelines, expression profiling revealed that 19 tumors (17.3%) had low baseline expression of HER2 and other genes in the amplicon (log10 ratio < −0.10) (Fig. 2b). Seventeen of these tumors with adequate remaining tissue were retested centrally by FISH, with 70.6% confirmed to be HER2 amplified and 29.4% found to be HER2 non-amplified. Of the low-HER2 expressing tumors, 84.2% (16/19) were HR+ and only 21.1% (4/19) had a pCR, suggesting that low HER2 expression in spite of HER2 amplification may correlate with lack of response.

Using the expression of genes beyond the hormone receptors and HER2 itself, breast cancer can be subcategorized into five intrinsic subtypes (including HER2-enriched) or into eleven integrative subtypes[36,37] (including IC5, representing the majority of HER2+ tumors). Intrinsic subtype has previously been shown to correlate with response to HER2-targeted therapy[26,27,29,30]. In this cohort, 56% of tumors were the HER2-enriched intrinsic subtype and 78% of tumors were IC5. For both subtyping approaches, the HER2 subtype trended toward a higher pCR rate than the non-HER2 subtypes: 50.0% vs 33.3% (OR 2.0 (95% CI 0.98–4.41)) for HER2-enriched vs other intrinsic subtypes; 47.7% vs 25.0% (OR 2.7 (95% CI 1.0–8.1)) for IC5 vs other integrative subtypes. The greater proportion of tumors classifying as IC5 versus HER2-enriched was driven by 23% of IC5 tumors classifying as the normal-like intrinsic subtype, which may encompass tumors with high non-tumor cell contamination[30,38], suggesting that integrative cluster assignment may be less sensitive to differences in tumor cellularity than intrinsic subtype

assignment. Indeed, tumor cellularity, defined as the proportion of evaluated cells representing invasive tumor cells, was found to be higher in HER2-enriched tumors than in normal-like tumors (17.7% vs 6.4%, two-sided t-test $p = 5.5e-6$), while no such difference was observed between IC5 and other integrative clusters (15.5% vs 13.0%, $p = 0.39$) (Supplementary Fig. 1).

Because HER2-enriched and IC5 tumors tended to be HR- (48.4% of HER2-enriched vs 33.3% of other, $\chi2\ p = 0.16$; 48.4% of IC5 vs 16.7% of other, $p = 0.0096$) and to have higher HER2 FISH ratios (mean HER2 FISH ratio 7.4 in HER2-enriched vs 5.8 in other, two-sided t-test $p = 0.050$; 7.5 in IC5 vs 4.2 in other, $p = 1.7e-5$), it is plausible that these clinically assessed variables dictate in part the association between tumor subtype and pCR. Our base multivariate model of pCR included nodal status, receipt of trastuzumab (Arms 1 and 3 vs Arm 2), HR-status, and HER2 FISH ratio (tumor size and patient age did not correlate with pCR in univariate models). Indeed, when intrinsic subtype or integrative subtype were added to this model, they were not significant while the original four variables remained so (Supplementary Table 3).

**Pre-treatment tumor and microenvironment gene expression signatures.** To assess variability between pre-treatment tumor gene expression profiles, we calculated single sample gene enrichment analysis (ssGSEA) scores[39] for 10 curated gene sets: tumor signaling gene sets (ESR1[40], ERBB2[40], and proliferation[41]), an immune gene set designed to capture immune infiltration in tumors (ESTIMATE)[42], immune gene sets identified as relevant to HER2-targeted therapy response (GeparSixto[33], Teff[43,44], and N9831[45]), a stromal gene set designed to capture stromal infil-tration in tumors (ESTIMATE)[42], and two stromal gene sets (Desmedt and Farmer) identified as relevant to breast tumors[40,46] (Fig. 2c). We similarly calculated ssGSEA scores for 38 of the

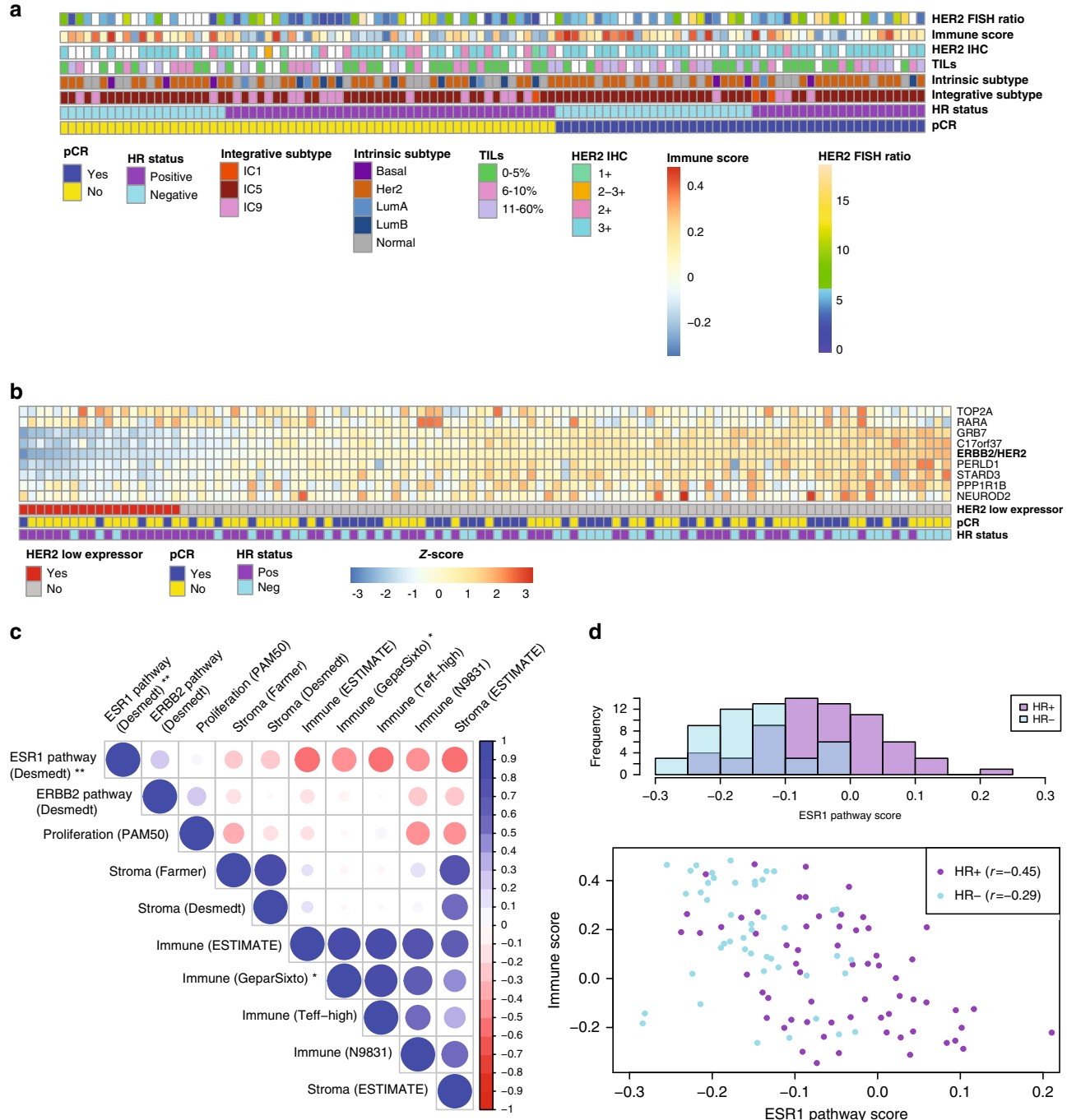

**Fig. 2 Characteristics of the cohort prior to treatment. a** Selected clinical and expression characteristics and tumor subtypes across the cohort. White squares reflect missing data. **b** Expression values of selected genes within the HER2 amplicon. **c** Pearson correlation coefficient matrix of key gene expression signatures. *$P = 0.010$ and **$P = 0.012$ for correlation with pCR (two-sided Wald test). **d** Distribution of ESR1 pathway gene expression scores (from ref. [39]) by HR subtype (top) and their correlation with immune scores (from ref. [33]) (bottom). FISH: fluorescent in situ hybridization; IHC: immunohistochemistry; TILs: tumor-infiltrating lymphocytes; HR: hormone receptor; pCR: pathologic complete response; IC; integrative subtype/integrative cluster.

Hallmark molecular signatures related to tumor processes and microenvironment[47] (Supplementary Fig. 2). The Hallmark signatures captured similar processes to the curated signatures, with several processes likely reflecting the change in microenvironmental composition: for example, the Hallmark epithelial-mesenchymal transition (EMT) signature correlated with the breast cancer stromal signatures ($r = 0.86$–$0.91$) and the Hallmark KRAS signaling signature correlated with the immune signature (ESTIMATE) ($r = 0.76$), suggesting each of these

captured a relative change in the number of stromal or immune cells rather than differences between the tumor cells.

As expected, the highly correlated estrogen signatures were associated with pCR, the strongest being the Hallmark estrogen response early signature (univariate logistic regression $p = 0.0017$), including within HR+ tumors separately ($N = 64$, $p = 0.012$). The ERBB2 pathway ($p = 0.38$) and proliferation ($p = 0.86$) signatures did not associate with pCR. Stromal signatures have been linked to therapeutic response in breast

cancer, including HER2+ disease[26,46], but in our cohort, there was no correlation between the stromal signatures and pCR. These results did not change when subsetted by treatment arm or by hormone receptor status. The breast cancer stromal signatures correlated well with each other (Pearson correlation coefficient $r = 0.85$) and less well with the general stromal signature ($r = 0.56–0.76$).

Immune gene sets were tightly correlated ($r = 0.81–0.90$ for the ESTIMATE, GeparSixto, and Teff). One immune signature (N9831) was more distinct from the other three ($r = 0.53–0.80$), and also anti-correlated with proliferation ($r = -0.46$), suggesting it may combine information about tumor proliferation and immune infiltration. A higher percentage of stromal tumor-infiltrating lymphocytes (TILs), which are dispersed in the stroma between the carcinoma cells and have been shown to predict therapeutic response in HER2+ breast cancer[32], was associated with a higher expression-based immune score (GeparSixto immune score 0.21 for >10% sTILs vs −0.054 for ≤5% sTILs, two-sided $t$-test $p = 1.2e-5$) (Supplementary Fig. 3). Tumors with >10% stromal TILs trended toward higher pCR rate (OR 2.3 (95% CI 0.9–6.3)), and similarly, tumors with higher immune scores trended toward higher pCR rate: the most associated was GeparSixto ($\beta = 1.5$ (95% CI −0.3–3.2)) (Supplementary Table 2). The trend toward higher immune infiltration predicting greater pCR was driven by the trastuzumab-containing arms, potentially consistent with a greater role of the immune system in response to the antibody trastuzumab than the small molecule inhibitor lapatinib[48] ($\beta = 3.2$ for TCH, $\beta = -0.5$ for TCL, and $\beta = 1.9$ for TCHL; interaction $p = 0.20$) (Supplementary Table 2). Importantly, immune gene signatures correlated strongly with HR-status, itself associated with pCR: using GeparSixto, the mean immune score was 0.17 in HR-negative versus 0.02 in HR-positive (two-sided $t$-test $p = 5.5e-4$). Interestingly, while we observed a trend toward increased immune infiltration correlating with pCR in the HR+ subgroup ($\beta = 2.1$ (95% CI −0.4–4.8)) (not in the HR- subgroup), we also found that the ESR1 pathway score[40] negatively correlated with immune infiltration among HR+ tumors (Fig. 2d). Since decreased HR signaling and increased immune infiltration correlate strongly with each other, even within HR+ tumors, either or both may contribute to increased response to HER2-targeted therapies.

We next examined the prevalence of immune cell subtypes before treatment using CIBERSORT[49] and immunoStates[50], which each use bulk gene expression data to deconvolve immune subpopulations. To limit multiple hypothesis testing, we examined the ten cell types with the largest mean prevalence across both deconvolution methods (plasma cells, B cells, CD8+ T-cells, CD4+ T-cells, NK cells, M0 macrophages, M1 macrophages, M2 macrophages, mast cells, and dendritic cells). While per histopathology, most of the inflammatory infiltrate was assessed to be lymphocytic (mean 91.8%), the predominant population(s) per CIBERSORT was macrophages (mean 48.8%) and per immunoStates were CD4+ T cells (mean 24.9%) and mast cells (mean 22.4%) (Supplementary Fig. 4A, B). Using CIBERSORT, CD8+ T-cell fraction was higher in HR- tumors (two-sided FDR-adjusted $t$-test $p = 0.045$), and no cell subtype correlated with pCR (Supplementary Fig. 4C, D). Using immunoStates, M1 macrophage fraction was higher in HR− tumors (adjusted $p = 0.048$) and NK cell fraction higher in HR+ tumors (adjusted $p = 0.0048$), and similarly no cell type correlated with pCR (Supplementary Fig. 4E, F). Across both immune deconvolution methods, with higher immune content, CD8+ T-cell (CIBERSORT: $r = 0.43$, FDR-adjusted $p = 3.2e-5$, immunoStates: $r = 0.27$, FDR-adjusted $p = 0.042$) and M1 macrophage (CIBERSORT: $r = 0.27$, FDR-adjusted $p = 0.47$, immunoStates: $r = 0.53$, FDR-adjusted $p = 2.39e-8$) fractions were higher. These immune

cell subtypes and total immune content correlated in the same direction when stratifying by HR-status (Supplementary Fig. 5). These results suggest that CD8+ T-cells and M1 (anti-tumor) macrophages may either infiltrate or be excluded from tumors separately from other immune cell subsets, and that similar temporal patterns of immune cell infiltration occur in HR− and HR+ tumors.

**Global gene expression changes after short-term HER2-targeted therapy**. We next examined the gene expression profiles of the 89 tumors collected 2–3 weeks after initiation of HER2-targeted therapy, comparing each tumor against its matched pre-treatment control. We observed dramatic changes in tumor phenotype with short-term targeted therapy at the level of expression-based subtype, gene signatures, and individual genes.

Histopathology-based mean tumor cellularity was estimated to be 14.7% pre-treatment and 6.5% on-treatment (two-sided paired $t$-test $p = 9.3e-9$) and, notably, there was no tumor present in 39% of the 83 on-treatment biopsies assessed. While the lack of cellularity was unlikely to reflect an absence of tumor tissue after a single cycle of HER2-targeted therapy, interestingly, an on-treatment biopsy without carcinoma was modestly predictive of pCR (OR 2.5 (95% CI 1.0–6.5)), suggesting biologically relevant patchiness of tumor occurring very early in treatment, though this variable was not significant when added to our base multivariate model of pCR (Supplementary Table 3), as absence of tumor was more common in the combination arm ($\chi2$ $p = 0.0079$).

Over half (53.9%) of tumors changed their intrinsic subtype after the short course of HER2-targeted therapy, with 79.2% of these converting to the normal-like subtype (Fig. 3a; Supplementary Figs. 6 and 7). Tumor subtype conversion to normal-like was driven at least in part by the reduced cellularity on-treatment: 87.0% of the biopsies with no identified tumor classified as normal-like, as compared to 56.7% of the biopsies with identified tumor ($\chi2$ $p = 0.026$) (Supplementary Fig. 8). Unlike what has been reported previously[30], conversion to normal-like was not associated with pCR (34.2% vs 33.3%), and conversion to luminal A was infrequent, suggesting that the addition of endocrine therapy may have driven this conversion in the previous study[35]. Integrative subtype was more stable than intrinsic subtype across targeted therapy, perhaps due to it being less dependent on microenvironmental composition: integrative subtype changed in 25.8% of tumors (vs 53.9% for intrinsic subtype; $\chi2$ $p = 2.4e-4$), with 82.6% of these converting to integrative subtype 9 (IC9, likely misclassification of lower cellularity tumors) (Fig. 3a; Supplementary Figs. 6 and 7). Similar to the intrinsic subtype, conversion to IC9 was not associated with pCR (31.6% vs 34.3%).

We evaluated the same signatures on-treatment as we did pre-treatment to generate a global picture of tumor signaling and microenvironmental change with therapy. Using gene set enrichment analysis (GSEA)[51], of 47 signatures (Teff-high excluded as only 3 genes), 20 signatures increased at FDR < 0.1, including stromal and immune signatures, and 15 decreased, including proliferation, ESR1 signaling, and ERBB2 signaling (Fig. 3b); these results were similar when analyzing only the subset of tumors with known tumor present (correlation coefficient of the normalized enrichment scores $r = 0.98$) (Supplementary Fig. 9). We then used ssGSEA scores[39] for each of these signatures to quantify the degree of change in each signature across treatment per tumor. Many of the changed signatures correlated strongly with each other (Fig. 3c), but three categories of change appeared to be largely distinct: a decrease in proliferation, an increase in immune component ($r = -0.19$ between PAM50 proliferation and immune ESTIMATE scores,

$p = 0.069$), and an increase in stromal component ($r = -0.15$ between Farmer stromal and PAM50 proliferation scores, $p = 0.17$; $r = 0.39$ between Farmer stromal and immune ESTI-MATE scores, $p = 1.2e-4$). While the various immune signatures correlated tightly (Fig. 3c), we used the immune ESTIMATE gene set going forward, as it was designed to capture overall immune infiltration in tumors in an unbiased way, unlike the GeparSixto and Teff-high gene sets, which were designed to predict response.

Both ERBB2 signaling and ESR1 signaling were anti-correlated with immune infiltration and stromal infiltration, making it difficult to disentangle whether their observed decreases were solely a result of a reduction in tumor content relative to other cell types (with less ERBB2 and ESR1 signaling), or whether the tumor cells independently experienced a reduction in the activation of these pathways in response to treatment. We constructed a linear model that predicted ERBB2 (or ESR1)

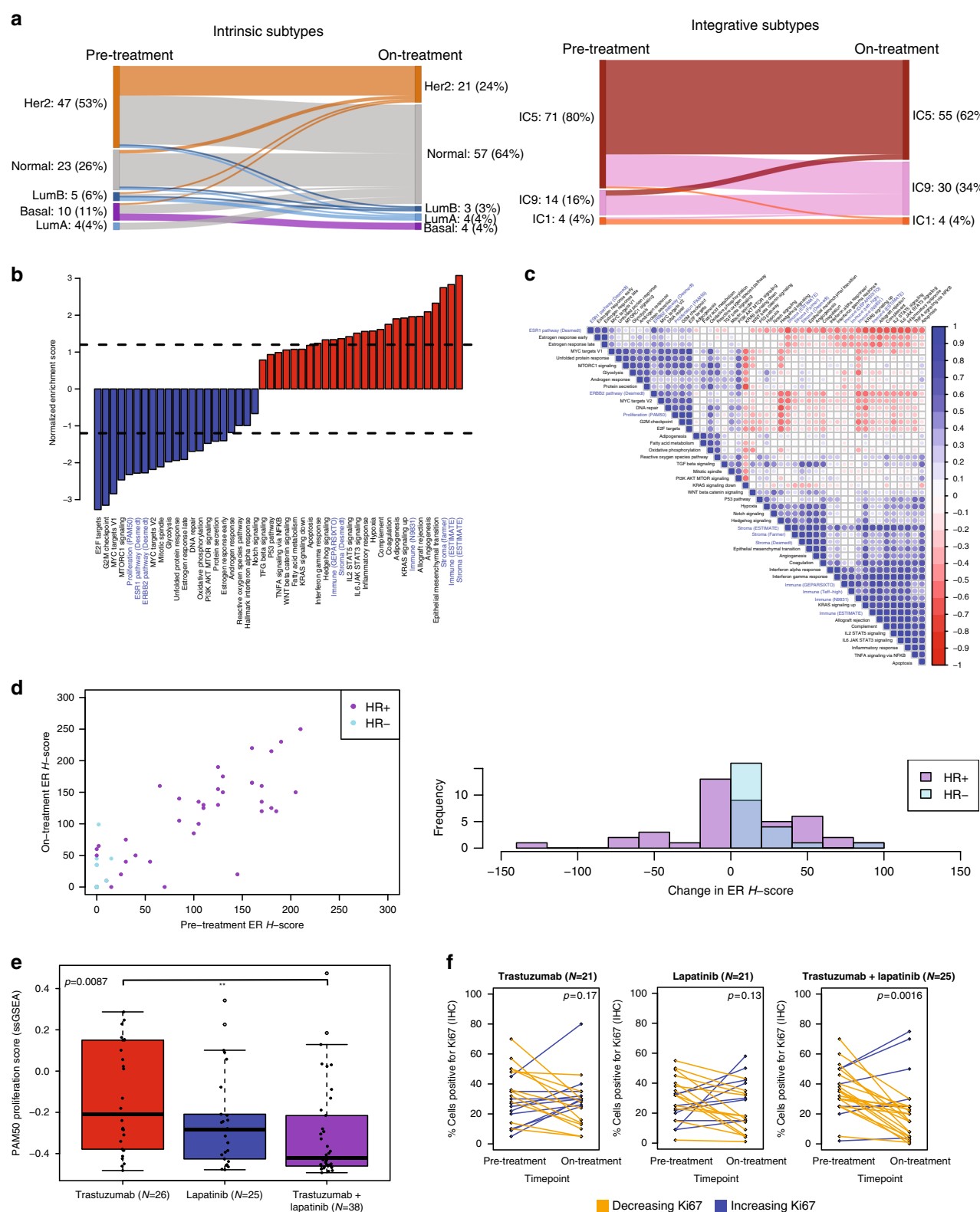

**Fig. 3 Tumor and microenvironmental changes on short-term HER2-targeted therapy. a** Subtype classifications (intrinsic and integrative) pre-treatment and after 14–21 days of HER2-targeted therapy. **b** Normalized enrichment scores from gene set enrichment analysis (GSEA) representing the degree of change of each gene set after 14–21 days of HER2-targeted therapy. Dotted lines separate those with FDR < 0.1. Black gene sets are Hallmark Molecular Signatures and blue gene sets were curated. **c** Pearson correlation coefficient matrix of single-sample GSEA (ssGSEA) scores. Ordering of gene sets is based on hierarchical clustering. Gene sets without labeled source in parentheses are Hallmark Molecular Signatures. **d** ER IHC H-score pre-treatment and after 14–21 days of HER2-targeted therapy correlate ($r = 0.83$) (left), and few tumors shift their ER H-score substantially (right) with treatment. **e** Change in proliferation ssGSEA scores after 14–21 days of HER2-targeted therapy by treatment arm. The mean drop in proliferation was highest with combination therapy, followed by lapatinib therapy (two-sided $t$ test $p = 0.28$ compared to combination therapy), followed by trastuzumab therapy ($p = 0.0087$ compared to combination therapy; $p = 0.13$ compared to lapatinib). Centerline is median, box limits are upper and lower quartiles, whiskers are 1.5x the interquartile range, and empty points are outliers. **f** Change in percentage of cells positive for Ki67 by immunohistochemistry, pre-treatment to after 14–21 days of HER2-targeted therapy, stratified by treatment arm. $P$-values are from two-sided paired $t$-tests of the log-transformed Ki67 values. HR: hormone receptor; ER: estrogen receptor; $H$-score: histochemical score.

signaling from immune ESTIMATE and Farmer stromal scores, finding that observed ERBB2 signaling correlated with predicted ERBB2 signaling ($r = 0.48$) less than observed ESR1 signaling correlated with predicted ESR1 signaling (0.69) (Supplementary Fig. 10). Using IHC to assess change in ER protein expression on therapy ($N = 73$), we did not observe a reduction in the histochemical score (H-score) on treatment (rather, there was a trend toward increased ER expression: 60.2 pre-treatment vs 69.5 on-treatment, two-sided paired $t$-test $p = 0.066$), and on-treatment H-score correlated closely with pre-treatment H-score ($r = 0.83$) (Fig. 3d). The IHC results suggest that indeed the apparent reduction in ESR1 signaling observed globally was a result of change in microenvironmental composition rather than changes in the tumor cells themselves. In contrast, in a parallel study, in situ proteomic analysis performed on this cohort indicated that ERBB2 protein levels in tumor cells did decrease after 14–21 days of HER2-targeted therapy[52]; thus, while increased immune and stromal components of the total sample likely contributed to the reduction in ERBB2 signaling seen in global gene expression, the tumor cells themselves also exhibited reduced ERBB2 expression.

None of the four major categories of change (proliferation, immune, stroma, and ERBB2 signaling) correlated with pCR (Supplementary Fig. 11), and in contrast to previous reports[53,54] as well as protein data from this cohort[52], on-treatment immune infiltration, whether assessed using immune score or stromal TILs, was no more predictive than pre-treatment immune infiltration (Supplementary Fig. 12). There were also no differences in the changes in signatures by HR status (Supplementary Fig. 13). However, two signatures were significantly different by arm (FDR < 0.1, adjusting for 12 hypotheses): the stromal signature increased more with L than with T (FDR-adjusted two-sided $t$-test $p = 0.078$), with TL intermediate, and the proliferation signature decreased more with TL than with T (FDR-adjusted $p = 0.078$), with L intermediate (Fig. 3e). To confirm that these changes occurred in the tumor cells, we examined Ki67 IHC, recapitulating that the greatest proliferation reduction was seen with TL, followed by L, followed by T: the geometric mean of Ki67 percentage changed across treatment from 29.8% to 11.4% with TL ($N = 25$; $p = 0.0016$), from 23.8% to 17.3% with L ($N = 21$; $p = 0.13$), and from 25.9% to 23.1% with T ($N = 21$; two-sided paired $t$-test of the log2 Ki67 values $p = 0.17$) (Fig. 3f).

Importantly, while proliferation and stromal signatures varied by treatment arm, immune signatures did not (Supplementary Fig. 11); given the lack of a control arm undergoing repeated biopsy without intervening treatment, it is impossible to be certain whether the immune changes observed were related to HER2-targeted therapy or repeated biopsy. Indeed, we note that hemoglobin subunits (HBA1, HBA2, and HBB) were among the top 1 percent of genes to increase from pre- to on-treatment, likely reflecting the impact of the biopsy. While the increase in

hemoglobin subunit expression did not correlate with the increase in immune expression ($r = 0.06$ between HBA1 expression and the immune ESTIMATE score), it remains possible that the immune changes observed were not related to the therapy itself.

**Microenvironmental changes across HER2-targeted therapy and chemotherapy.** A total of 59 tumor or tumor bed samples were collected at surgery after completion of combination chemotherapy and HER2-targeted therapy, and as with the samples after 14–21 days of HER2-targeted therapy, the gene expression of each was compared against its matched pre-treatment tissue. Of these 59 surgical samples, 25 tumors had undergone pCR with no tumor remaining. For a subset ($N = 39$), histopathology from the region of the surgical resection used for gene expression analysis was re-assessed centrally, and here 73% of even the non-pCR cases showed no evidence of tumor in the analyzed sample. Thus, it was not possible to assess tumor changes after combination chemotherapy and HER2-targeted therapy, and we focused our analyses on characterizing the immune and stromal changes observed in the tumor bed.

Evaluating the same 47 gene sets at the time of surgery compared to pre-treatment by GSEA (Fig. 4a), reductions in proliferation, ERBB2 signaling, and ESR1 signaling were observed, consistent with lower tumor content. Overall, the correlation between the GSEA normalized enrichment scores after HER2-targeted therapy and after combination chemotherapy and HER2-targeted therapy was $r = 0.68$. The major differences were in stromal and immune signatures (Fig. 4b). The four stromal signatures (ESTIMATE[42], Hallmark EMT, and the two breast cancer-specific stromal signatures[40,46]) increased synchronously with HER2-targeted therapy alone; after chemotherapy, however, the non-breast cancer signatures remained elevated, while the breast cancer stromal signatures plummeted, perhaps because these signatures capture gene expression related to stromal interaction with the active tumor or that is affected differentially by chemotherapy.

The immune signatures followed a similar pattern to the breast cancer-specific stromal signatures, increasing after HER2-targeted therapy alone, but decreasing after chemotherapy in combination with HER2-targeted therapy. Importantly, this result differed from in situ proteomic analysis on this cohort performed in a parallel study[52], where immune cells increased at surgery compared to pre-treatment. This discrepancy is likely explained by the very low, often zero, tumor cellularity in the surgical samples on which bulk expression was assessed, as contrasted with the tumor enrichment strategy used for in situ proteomic analysis of a separate formalin-fixed paraffin-embedded core: it is plausible that the larger tumor bed tissue is largely an immune desert, while localized immune cell infiltration continues to occur proximal to tumor cell islands. Indeed, the histopathologic data supports this hypothesis (Fig. 4c). Stromal TILs, which by

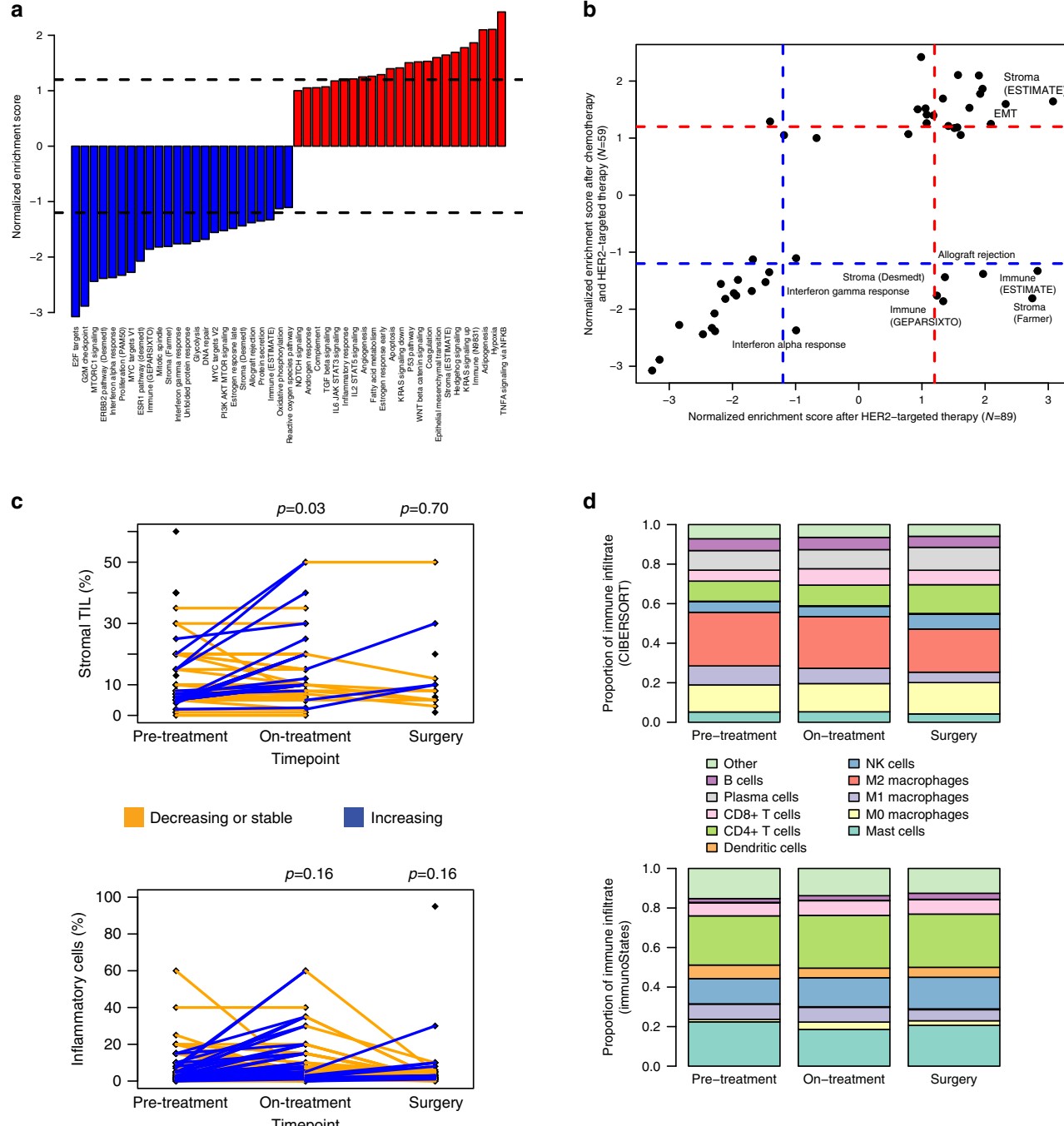

**Fig. 4 Microenvironment changes across HER2-targeted therapy and chemotherapy. a** Normalized enrichment scores from gene set enrichment analysis (GSEA) representing the degree of change of each gene set at the time of surgery compared to pre-treatment. Dotted lines separate those with FDR < 0.1. **b** Normalized enrichment scores at surgery compared to pre-treatment (y-axis) vs after 14–21 days of HER2-targeted therapy compared to pre-treatment (x-axis). Dotted lines separate those with FDR < 0.1. Immune and some stromal sets increase after 14–21 days of HER2-targeted therapy but are reduced at the time of surgery. **c** Change in stromal TILs and inflammatory cell percentage across treatment. *P*-values are from two-sided paired *t*-tests, comparing on-treatment to pre-treatment (N = 55 for stromal TIL and N = 94 for inflammatory cells) and surgery to pre-treatment (N = 14 for stromal TIL and N = 78 for inflammatory cells). **d** Proportion of immune infiltrate represented by each immune cell subtype, according to CIBERSORT (top) and immunoStates (bottom). Pre-treatment (N = 89) is matched to on-treatment (N = 89); surgery (N = 59) is a subset. TIL: tumor-infiltrating lymphocytes.

definition are tumor-adjacent, increased modestly after 14–21 days of HER2-targeted therapy (from 9.9% to 12.7%, two-sided paired *t*-test *p* = 0.034), and, in the 14 tumors with non-zero cellularity samples both pre-treatment and at time of surgery, were stable after combination chemotherapy and HER2-targeted therapy (8.5% vs 9.5%, two-sided paired *t*-test *p* = 0.70), consistent with continued immune infiltration occurring where

the tumor was present. When we assessed for inflammatory cellularity across the entire slide, including samples with no tumor present, trends were consistent with the expression data: increased inflammatory cellularity on-treatment (from 6.1% to 8.0%, two-sided paired *t*-test *p* = 0.16) and reduced inflammatory cellularity at surgery (from 5.9% to 3.5%, two-sided paired *t*-test *p* = 0.16).

We used CIBERSORT[49] to deconvolve the components of the immune infiltrate across HER2-targeted therapy and chemotherapy (Fig. 4d). CIBERSORT absolute scores increased from pre-treatment (mean 0.55) to on-treatment (0.60; two-sided paired $t$-test $p = 0.011$) and returned to baseline at surgery (mean 0.54; $p = 0.53$ vs pre-treatment). After 14–21 days of HER2-targeted therapy, the three most significant changes compared to pre-treatment were a reduction in plasma cells (14.8% to 9.7%, two-sided paired $t$-test FDR-adjusted $p = 6.6$e-8), a reduction in M1 macrophages (10.2% to 7.7%, FDR-adjusted $p = 1.3$e-5), and an increase in CD8 + T-cells (5.2% to 8.3%, FDR-adjusted $p$-2.5e-4). At time of surgery after combination chemotherapy and HER2-targeted therapy, the two most significant changes compared to pre-treatment were a reduction in M1 macrophages (11.0% to 5.2%, FDR-adjusted $p = 5.0$e-11) and an increase in NK cells (4.6% to 7.4%, FDR-adjusted $p = 9.4$e-8). While the proportions of each cell type were substantially different with an alternative immune deconvolution approach, immunoStates[50] (Fig. 4d), again, after HER2-targeted therapy, CD8 + T-cells were observed to increase (FDR-adjusted $p = 1.2$e-9) and, after combination chemotherapy and HER2-targeted therapy, M1 macrophages to decrease (FDR-adjusted $p = 5.0$e-4) and NK cells to increase (FDR-adjusted $p = 2.8$e-7).

## Discussion

In this clinical trial, lapatinib plus trastuzumab did not improve pCR rate compared to trastuzumab alone when added to chemotherapy in the neoadjuvant setting, while the omission of trastuzumab yielded lower pCR rates (TCL 25% compared to TCH 47% and TCHL 52%). Other trials have also shown that the anti-tumor activity of trastuzumab is superior to that of lapatinib[20,21,27,55,56]. The benefit of combining trastuzumab and lapatinib is less clear. While NeoALTTO[20] showed a significantly improved pCR rate with 12 weeks of paclitaxel plus lapatinib and trastuzumab compared to one HER2-targeted agent plus paclitaxel, NSABP B-41[21] and CALGB 40601[27] did not show such an improvement with dual-HER2 targeted therapy compared to trastuzumab plus chemotherapy. Some of this discrepancy may be explained by the shorter course of chemotherapy with NeoALTTO compared to the other studies. However, each of these studies did show a numeric increase in pCR rate with lapatinib and trastuzumab compared to single-agent HER2-targeted therapy, while in this study, the rates were very similar when comparing TCH (47%) and TCHL (52%). Notably, in TRIO B07, a higher percentage of participants enrolled in TCH (100%) were able to complete all protocol-specified therapy prior to surgery compared to participants in the lapatinib-containing arms (69% in TCL and 74% in TCHL). In addition, the average relative dose intensity of therapy delivered to participants on TCH was higher (98%) compared to those treated on the lapatinib-containing arms (85% TCL and 86% TCHL). The dose reductions in the lapatinib-containing arms were necessitated by lapatinib-related toxicities and may have contributed to the decreased anti-tumor activity measured at the time of surgery.

In TRIO B07, whenever possible, both fresh-frozen and formalin-fixed paraffin-embedded tissue specimens were collected at three timepoints: pre-treatment, after 14–21 days of HER2-targeted therapy alone, and at surgery after completion of combination chemotherapy with HER2-targeted therapy. These serial specimens from the same patient permitted evaluation of multiple potential biomarkers of tumor sensitivity and resistance, as well as how these biomarkers relate to one another and how they change throughout therapy. As in previous studies[26,27,29,30], we find that tumors classified as the HER2-enriched intrinsic subtype have a trend toward higher pCR rate compared to other intrinsic subtypes. In this study, we also assessed integrative subtype[36,37], and similarly find a higher rate of pCR in IC5 as compared to the other integrative subtypes. The association of these breast cancer subtypes with response appears to be mediated, at least in part, by higher HER2 amplification and decreased HR expression in the HER2-enriched and IC5 subgroups, consistent with another report that intrinsic subtype was no longer associated with pCR when ESR1 and ERBB2 expression were included in a multivariate analysis[27]. Among HER2+ tumors, HER2 amplification levels may not help predict which tumors will benefit from trastuzumab[57,58], but it is plausible that it may help identify tumors that are very trastuzumab-sensitive, thus informing de-escalation approaches. Measures of pre-treatment immune infiltration, closely tied to HR signaling, may add some value to HR status and degree of HER2 amplification in predicting pCR, especially in HR+ tumors and especially with trastuzumab treatment (rather than lapatinib).

We identified numerous expression changes across HER2-targeted therapy including decreased proliferation, increased immune cell infiltration, increased stromal signatures, and decreased ERBB2 signaling. We similarly observed a high rate of change of intrinsic subtype across therapy, generally to normal-like, while integrative subtype was more stable. Interestingly, proliferation was the least reduced with trastuzumab alone and the most with dual HER2-directed therapy. The observation that lapatinib may be more effective than trastuzumab at suppressing proliferation has important implications for window of opportunity studies, where tumor proliferation is examined after short-term therapy to assess efficacy. Window of opportunity studies have largely been used to assess endocrine therapy, but are increasingly being applied to other targeted therapies as well[59–62]. Our results indicate that, when comparing two therapies, it is possible that the one that is more effective in terms of pCR and survival (here, trastuzumab) may actually induce an equivalent or lesser reduction in proliferation. Indeed, it is plausible that the greater proliferation reduction with lapatinib renders the cells less sensitive to the effects of chemotherapy, reducing pCR rates. It is also plausible that lapatinib induces more immediate changes in the tumor cells, but trastuzumab renders them more immunogenic. Notably, we observed a greater increase in stromal signal with lapatinib compared to trastuzumab, potentially consistent with lapatinib's unique interactions with the stromal compartment[63], but immune signatures increased similarly across all arms. Because this study did not include a control arm where participants underwent on-treatment biopsy without intervening therapy, we cannot rule out that the observed on-treatment immune infiltration related to repeated biopsy[64] rather than HER2-targeted therapy itself. However, we note that differences in the degree of immune infiltration observed were associated with pCR in the parallel in situ proteomic study[52], suggesting that on-treatment immune infiltration was of biological significance.

The increase in immune signatures, especially in CD8-positive T-cells, that we observed after 14–21 days of HER2-targeted therapy was eliminated by time of surgery, likely because the sample at surgery was largely devoid of tumor, as this result contrasts to what was observed with tumor enrichment and in situ proteomic profiling[52]. Combination chemotherapy and HER2-targeted therapy was also associated with a relative increase in NK cells, similar to what was observed in the proteomic data[52], and a relative decrease in M1-like (anti-tumor) macrophages. Whether the loss of the M1 macrophage signature was related to the tumor cells themselves, where parainflammation may resemble macrophage infiltration[65], or to a shift in macrophage phenotype toward a pro-tumor state as suggested with chemotherapy in HR+ /HER2– breast cancers[66], remains to be determined. These immune cell subset changes across therapy

may have implications for optimal timing of immune interventions, with manipulations that affect CD8-positive T-cell activity perhaps most useful with HER2-targeted therapy prior to chemotherapy.

Of the many expression changes we observed in tumors after 14–21 days of HER2-targeted therapy, none predicted pCR, in contrast to in situ proteomic profiling on this same cohort[52]. Notably, many of the on-treatment tumor specimens had no or very low tumor cellularity, reflecting real-world biopsy conditions but potentially obscuring important biological differences. Additionally, intra-tumor heterogeneity in response and poor correlation of RNA signatures with tumor phenotype could contribute to bulk RNA profiling of a single biopsy being a suboptimal approach to quantitatively compare changes across therapy. Thus, while our work identifies the changes across treatment that occur with HER2-targeted therapy with and without chemotherapy, additional approaches are needed to discern relevant biological variability in these changes that may predict therapeutic sensitivity or resistance. The uniform collection of tissue in this trial – both fresh-frozen and formalin-fixed paraffin-embedded – allows these questions to be explored further using a variety of novel technologies. Future studies would also benefit from longitudinal sampling, including on-treatment biopsies, as well as uniform tissue collection and storage such that iterative learning is possible from the initial characterization.

## Methods

**Patients**. Participants were recruited at the University of California, Los Angeles (UCLA) (including satellite clinics at Santa Monica, Valencia, UCLA Westlake, and Pasadena) and twelve additional United States sites through the Translational Research In Oncology (TRIO)-US network (Bakersfield, Fullerton, Redondo Beach, Inland Valleys, Santa Maria, Orlando (Florida), Santa Barbara (2 sites), Las Vegas, Olive View, Hollywood (Florida), and San Luis Obispo). Women age 18 to 70 were eligible if they had an ECOG performance status of ≤1 and anatomic stage I-III unilateral HER2-positive breast carcinoma. Two women older than 70 (ages 76 and 78) were enrolled with protocol exceptions. HER2 status was defined by locally assessed in situ hybridization (FISH or SISH) assays, and 2007 ASCO-CAP guidelines were used, requiring HER2 ratio to CEP17 > 2.2 or average HER2 copy number > 6 signals/nucleus. Of the tumors with non-missing information regarding type of ISH assay ($N = 116$), 89.7% used FISH and 10.3% used SISH. Participants with stage I disease were required to have a tumor size ≥ 1 cm and be either younger than 36 years, have a tumor grade ≥ 2, or have a hormone receptor-negative (HR-) tumor. Inflammatory breast cancer was allowed. Adequate renal, hematologic, hepatic, and cardiac function (left ventricular ejection fraction (LVEF) ≥ lower limits of normal) were required.

Exclusion criteria included prior exposure to chemotherapy, radiation, or endocrine therapy for currently diagnosed invasive or non-invasive breast cancer, any prior radiation therapy to ipsilateral breast or chest wall, history of any other malignancy within the past 5 years (except non-melanoma skin cancer or carcinoma-in-situ of the cervix), pre-existing motor or sensory neuropathy of grade ≥2, pre-existing cardiac disease, gastrointestinal condition causing chronic diarrhea requiring active therapy, concurrent infection requiring parenteral antibiotics, metastatic breast cancer, current treatment with ovarian hormonal replacement therapy, or current treatment with any selective estrogen receptor modulators. Pregnant or lactating women were excluded and contraception was required for females of childbearing potential.

The following institutional review boards approved the study protocol: UCLA, Olive View, and Western. The Stanford University institutional review board also approved the molecular analyses. The study was conducted following Good Clinical Practice guidelines and local laws and regulations. Prior to the performance of any study-related procedures, each participant signed an institutional review board-approved informed consent form. This trial was registered as NCT00769470 at www.clinicaltrials.gov.

**Study design, treatment, and assessments**. TRIO B07 was a randomized, multicenter, open-label, three-arm phase II study conducted by the Translational Research In Oncology (TRIO) study group. Participants were stratified based on baseline tumor size (≤3 cm and >3 cm) and hormone receptor status (HR+ vs HR−). A random permuted block design was utilized for randomization, with the block size varied between 3 and 6 at random. The study statistician generated the random allocation sequence. TRIO Liaison Coordinators enrolled the participants and assigned them to the intervention arms. The study included three treatment groups (Arms 1–3). All participants received docetaxel plus carboplatin (TC) every 3 weeks. In addition, participants in Arm 1 received trastuzumab (TCH), Arm 2

received lapatinib (TCL), and Arm 3 received both trastuzumab and lapatinib (TCHL). Eligible participants were treated initially with a run-in cycle of HER2-targeted therapy without chemotherapy (lapatinib at a dose of 1000 mg per day orally for 21 days and/or trastuzumab 8 mg/kg IV once) followed by six cycles of the assigned HER2-targeted treatment plus docetaxel and carboplatin given every three weeks. The six cycles of chemotherapy consisted of concomitant docetaxel (75 mg/m² IV) and carboplatin (area under the plasma concentration-time curve [AUC] 6 mg/mL/min) plus trastuzumab (6 mg/kg IV) or lapatinib (1000 mg/day orally days 1–21) or both trastuzumab and lapatinib at the respective doses. Participants were required to receive primary prophylactic white cell growth factors after each chemotherapy cycle. Treatment was discontinued upon completion of all prescribed protocol therapy, disease progression, unacceptable toxicity necessitating the discontinuation of study drug, or withdrawal of participant consent. Safety assessments were conducted throughout the study from day 1 through the end of study treatment visit, 28 days after surgery or from the time of study discontinuation. Toxicity was graded using the NCI Common Toxicity Criteria for Adverse Events (CTCAE), version 3.0. No more than seven cycles of trastuzumab and/or lapatinib (run-in and six cycles with chemotherapy) and no more than six cycles of TC chemotherapy were to be given prior to surgery. After the sixth cycle of chemotherapy, participants deemed to be surgical candidates proceeded with standard of care breast surgery and axillary lymph node sampling. We note that in an open-label clinical trial, there is the potential for selection bias as participants may elect drop out if randomized to an arm they perceive to be associated with less benefit. This could cause imbalance in measured/unmeasured baseline characteristics of patients in the treatment arms. Another potential source of bias is early discontinuation in an arm due to toxicity of the regimen. This could affect the pCR rate as well as correlative biomarker analyses.

To test the safety of TCHL, the first 20 participants enrolled were assigned to Arm 3. A safety analysis of the combination therapy was performed after these first 20 participants prior to opening up the expansion. The remaining participants were randomized evenly (1:1:1) to the three arms. The first 6 participants took part in a dose escalation evaluation of carboplatin (first 3 participants treated at an AUC of 5 mg/mL/min, next 3 treated at AUC 6 mg/mL/min) plus docetaxel (75 mg/m²), trastuzumab (6 mg/kg), and lapatinib (1000 mg/day). Docetaxel and carboplatin were infused as per institutional practice with standard steroid and anti-emetic prophylaxis. Treatment could be delayed up to 21 days for toxicity. When a cycle was held for toxicity, all drugs were held to maintain concurrent dosing. Dose delays and reductions were permitted for carboplatin (to AUC 5) and docetaxel (to 60 mg/m²) for toxicity including hematologic, hepatic, nervous system, and gastrointestinal toxicity. Dose reductions were not permitted for trastuzumab; however, dose delays were permitted. Dose delays and reductions (to 750 mg/day) were permitted for lapatinib-related diarrhea and moderate to severe cutaneous reactions.

All participants had an evaluation of cardiac function including measurement of left ventricular ejection fraction (LVEF) by either multiple gated acquisition (MUGA) or echocardiogram at baseline, after cycle 3 of chemotherapy, <28 days of surgery, and as needed. Treatment with trastuzumab and/or lapatinib was to be permanently stopped and the participant taken off study in cases of symptomatic congestive heart failure (CHF) or a significant drop in LVEF (>10 points below baseline LVEF and below institutional lower limit of normal) confirmed by a second LVEF assessment within approximately three weeks.

End of study visit occurred 28 days after surgery or from the time of study withdrawal/discontinuation for any reason. After coming off study, all participants were allowed to receive therapy according to standard of care (radiation, endocrine therapy, maintenance trastuzumab) at the discretion of their treating physician. Clinical tumor assessments by physical examination were performed at baseline, before each cycle of therapy, and at the completion of all prescribed protocol therapy. Radiologic assessments to exclude macrometastases were performed if clinically indicated.

**Endpoints and statistical analysis**. The primary objective was to investigate the clinical efficacy of each arm by estimating the total pCR rate defined as the absence of viable invasive tumor cells in both the breast and axillary lymph nodes at the time of definitive surgery (ypT0/is ypN0). The tumor pCR rate was determined for both the intent to treat (ITT) population (including all participants who received a dose of study drug, regardless of whether they completed all protocol-specified therapy) and for the evaluable participant population (including only those participants who completed the protocol-specified pre-surgical therapy and underwent definitive breast surgery). Pathology reports from definitive surgery were centrally reviewed for all participants enrolled on study who received at least one dose of study drug regardless of whether the participant completed protocol-specified therapy.

Based on the literature, the pCR rate was estimated to be 40% for single biological agent treatment (trastuzumab) combined with multi-agent chemotherapy[11,67–69]. A sample size of 56 participants was required to detect an absolute 20% difference in the pCR rate between the experimental treatment (with hypothesized 60% pCR rate) and the historical-control pCR rate (of approximately 40%) with a nominal one-sided 0.05 significance and 90% power using the exact binomial method[70]. Rates of pCR were reported with 95% confidence intervals using the prop.test function in R.

As a secondary efficacy analysis, the pCR rates were to be compared between the arms using Chi-square tests in a pairwise setting with a two-sided type I error rate of 20%. With 60 participants in the combination arm and 40 participants in the single biological arm, this test would have 75% power for testing a difference in the pCR rates of 40% vs. 60% between the two arms (TCH or TCL vs TCHL)[71]. Another exploratory pairwise comparison of combined groups TCH and TCL vs TCHL would be performed in a similar fashion.

Secondary endpoints included safety and tolerability including rate of CHF or a significant drop in LVEF (>10% points from baseline and below the institutional lower limits of normal) for each of the three treatment arms. The safety analysis was conducted on all participants who received all or any portion of one infusion of any study drug. Descriptive statistics were used to summarize the number and types of adverse events. Adverse events were compared using chi-squared ($\chi^2$) tests.

Because the first 20 women were enrolled to Arm 3 rather than being randomized, exploratory analyses of pathologic complete response (pCR) rates were also run excluding these 20. For these analyses, the overall pCR rate was again 43% (46/108), and for Arm 3 it was 55% (21/38) versus 52% when the first 20 were included. Rerunning the comparisons again revealed that pCR was significantly lower in Arm 2 compared to Arm 3 ($p = 0.01$), and no significant differences were found between Arms 1 and 2 ($p = 0.07$) or between Arms 1 and 3 ($p = 0.47$).

When running the prespecified analyses for the evaluable participant population, the pCR for Arms 1, 2, and 3 were 47% (16/34), 28% (7/25), and 49% (21/43), respectively. For this set of analyses, none of the pair-wise comparisons using Chi-square were significant. Using logistic regression adjusting for the stratifying factors of HR status and tumor size (≤3 cm versus >3 cm), however, yielded a significant result when comparing Arm 2 to Arm 3 ($p = 0.04$), but not for Arm 1 to Arm 2 ($p = 0.10$) or Arm 1 to Arm 3 ($p = 0.88$).

**Gene expression profiling.** Tumor tissue was obtained by core biopsy prior to the administration of the run-in cycle of lapatinib and/or trastuzumab (pre-treatment samples) and 14–21 days after the start of the run-in cycle (on-treatment samples). A minimum of 4 core biopsies, using a 14-gauge needle, were taken at each timepoint, three of which were immediately snap-frozen and the fourth was formalin-fixed and paraffin embedded (FFPE). RNA was extracted using the RNeasy Mini Kit (Qiagen, Valencia, CA, USA), quantified by the Nanodrop One Spectrophotometer (ThermoFisher Scientific, Wilmington, DE, USA) and quality (RNA Integrity Number, RIN value ≥ 5) was confirmed by capillary electrophoresis using the Bioanalyzer 2100 (Agilent Technologies, Santa Clara, CA, USA). RNA samples were labeled with cyanine 5-CTP or cyanine 3-CTP (Perkin Elmer, Boston, MA, USA) using the Quick AMP Labeling Kit (Agilent Technologies). Labeled RNA was purified on RNeasy columns (Qiagen). Frequency of incorporation and total concentration of labeled RNA were determined using the Nanodrop One Spectrophotometer. In each case, 825 ng of a cyanine 5-CTP and a cyanine 3-CTP labeled RNA were applied to each slide. Slides were incubated for 16–17 hr at 65 °C. The slides were washed using wash buffer provided from Agilent then covered by ozone barrier as described in Agilent 60-mer oligo microarray processing protocol. Slides were read using the Agilent Scanner (G2565CA), and the data were extracted using Agilent Feature Extraction Software (versions 10.7 and 11.0). Gene expression microarray experiments were performed using the Agilent Whole Human Genome 44 K 2-color chip by comparing each baseline sample to a reference mix composed of 19 breast tumors samples which include HR-positive, HER2-amplified, and triple-negative samples (pre-treatment), by comparing each baseline sample to its matched sample taken after 2–3 weeks of treatment with HER2-targeted therapy (on-treatment), and by comparing each baseline sample to its matched sample taken at time of definitive surgery (post-treatment).

**Histopathology.** Tumor sections stained with hematoxylin and eosin were evaluated by a board-certified breast pathologist (GRB) blinded to clinical and response information. The following measures were assessed: tumor cellularity, inflammatory cellularity (defined as the percentage of all cells over the slide estimated to represent inflammatory cells), stromal tumor-infiltrating lymphocytes (TILs) according to the International TILs Working Group System[72], and percentage of inflammatory cells estimated to represent lymphocytes.

**Immunohistochemistry.** Ki67 (DAKO M7240, dilution 1:200) and ERα (Agilent SP1, M3634, dilution 1:200) immunohistochemistry (IHC) were performed on tumors pre-treatment and after 14–21 days of therapy. Unstained sections were immuno-stained according to previously described procedures[73] and scored blindly by a board-certified breast pathology (MFP). IHC images were digitized on the Aperio Digital Pathology Slide Scanner, and Aperio ImageScope software (version 12.3.2.8013) was used for image visualization and acquisition. For Ki67 percentage, the proportion positive reflects the percentage of tumor cells qualitatively scored with intensities as strong (3+), moderate (2+), and weak (1+). For ER, the histochemical score (H-score) represents the sum of 3 x the percentage of strongly staining (3+) nuclei, 2 x the percentage of moderately staining (2+) nuclei, and the percentage of weakly staining (1+) nuclei.

**Expression analyses.** Limma (version 3.28.21) was used for background correction (normexp), within-array normalization (loess), and between-array

normalization (for single-channel analyses only)[74,75]. ComBat (sva version 3.20.0) was used to remove potential batch effects associated with microarray run date[76]. Using single-channel data, PAM50 intrinsic subtype was predicted using Absolute Intrinsic Molecular Subtyping (AIMS version 1.4.0)[77] and integrative subtype using the iC10 classifier (genefu version 2.16.0), excluding the normalizeFeatures step given uneven subtype distribution within the cohort[37]. Immune cell populations were quantified using CIBERSORT version 1.06[49] and immunoStates (MetaIntegrator version 2.1.1)[50]. Immune composition of the sample was quantified using ESTIMATE version 1.0.13[42]. Using dual-channel data, gene set enrichment analysis (GSEA)[51] (GSEA version 4.0.2) was used to assess signature changes across treatment and single-sample GSEA (ssGSEA)[39] to compare individual gene signature scores between tumors (GSVA package version 1.32.0 and GSEABase package version 1.46.0). The following 12 Hallmark signatures were not evaluated given lack of relationship to tumor processes or microenvironment: apical surface, apical junction, peroxisome, pancreas beta cells, spermatogenesis, bile acid metabolism, heme metabolism, cholesterol homeostasis, UV response up, UV response down, xenobiotic metabolism, myogenesis.

**Statistics and reproducibility.** Pearson correlation coefficients were calculated between gene signatures. Differences between gene signatures by baseline tumor characteristics such as hormone receptor status or tumor subtype were assessed with two-sided $t$-tests. Differences between gene signatures, Ki67 or ER immunohistochemistry, and immune composition across time were assessed with two-sided paired $t$-tests. Differences between proportions by baseline tumor characteristics were assessed with $\chi^2$ tests. The associations between expression-based subtype, gene signatures, and other variables with pCR were assessed using logistic regression, with two-sided Wald tests of the null hypothesis that the coefficient is equal to zero used to report $p$-values and with the confint function in R used to report confidence intervals of estimates. R versions 3.5.1 and 3.6.1 were used for all statistical analyses. The exact numbers of samples used for each comparison made of the gene expression data are reported in the text and figure legends, with the maximum being $N = 110$ for pre-treatment samples, $N = 89$ for on-treatment samples, and $N = 59$ for surgical samples; however, some comparisons making use of histopathologic data (immunohistochemistry, stromal tumor infiltrating lymphocyte percentage, fluorescent in situ hybridization data) have smaller sample sizes.

**Reporting Summary.** Further information on research design is available in the Nature Research Reporting Summary linked to this article.

## Data availability
Raw expression data are deposited in GEO (GSE130788). The full trial protocol and processed data associated with the manuscript are available at github.com/cancersysbio/TRIOB07/. The processed data files include: preTreatment.txt (Figs. 2, 4c, d, Supplementary Figs. 1–5, 13), on Treatment.txt (Figs. 3a, c–f, 4c, d, Supplementary Figs. 6–9, S11–S14), postTreatment.txt (4c, d), and GSEA.txt (Figs. 3b, 4a, b, Supplementary Fig. 10).

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

## Acknowledgments

We would like to thank the women who participated in the trial, Jeffrey Miller for his role as site principal investigator, He-Jing Wang for statistical support, Susie Brain, Diane Heditsian, and Vivian Lee for helpful discussions, and the UCLA Translational Pathology Core Laboratory and UCLA/JCCC Translational Oncology Research Laboratory (especially Enrique Guandique and Lillian Ramos) for pathology support. Funding for the clinical trial came from grants from Sanofi aventis and GlaxoSmithKline. S.H. was supported in part by NCI/NIH CA016042 as well as the Marni Levine Memorial Research Award. J.L.C. was a Damon Runyon Physician-Scientist supported by the Damon Runyon Cancer Research Foundation (PST 11-17) and a Susan G. Komen Postdoctoral Fellowship Award (PDR17481769). K.M. was supported by an award from the National Cancer Institute at the National Institutes of Health (F30CA239313-02). Funding for the molecular analyses was supported in part by an award from the National Cancer Institute at the National Institutes of Health (R01CA182514) to C.C. and an award from the Breast Cancer Research Foundation to C.C. Funding for the immuno-histochemistry analysis was supported in part by a Department of Defense Breast Cancer Research Fellowship Award, Award Number W81XWH-11-1-0572 (J.Z.) and by a StandUp To Cancer-American Association for Cancer Research Dream Team Translational Cancer Research Grant, Grant Number SU2C-AACR-DT0409 (J.B.). The funding sources had no role in the design of the study nor any role in the analysis and interpretation of data or decision to submit results.

## Author contributions

S.H. designed the study and served as PI of the trial. J.C., K.M., J.D., H.W.C., and E.K. analyzed molecular data. J.Z., M.P., and J.B. performed immunohistochemistry assays. M.P. performed central FISH testing for a subset of cases. G.B. performed histopathological analysis. R.D., A.P., R.P., L.Z., H.A., L.B., B.D., A.K., A.G., C.Ca., D.M., and A.M. served as site PIs of the trial. B.A. was the project manager for the trial. D.S. supervised the trial design and execution. C.C. supervised the molecular and outcome association analyses. S.H., J.C., and C.C. wrote the manuscript, which was reviewed by all authors.

## Competing interests

S.H. received contracted research and medical writing assistance from Ambrx, Amgen, Arvinas, Bayer, Daiichi-Sankyo, Genentech/Roche, GSK, Immunomedics, Lilly, Macrogenics, Novartis, Pfizer, OBI Pharma, Pieris, Puma, Radius, Sanofi, Seattle Genetics, and Dignitana. A.P. received institutional research support from Genentech, AstraZeneca, Immunomedics, Nektar, and Macrogenics. L.B. received honoraria from and/or was on the speaker bureau for AstraZeneca, Pfizer, Merck, Puma, Novartis, GlaxoSmithKline, Amgen, Johnson & Johnson, Genentech/Roche, Sandoz, Dendreon, and Genomic Health, and performed consulting work for Integra Connect and Anthem Blue Cross. A.M. is employed by Genentech/Roche and owns stock in Roche. M.P. received research support from Cepheid, was on the Scientific Advisory Board and received research support from Eli Lilly, Zymeworks, and Novartis, was a consultant for and received research support from Puma, and was on the Scientific Advisory Board for Biocartis. C.C. is a scientific advisor to GRAIL and reports stock options as well as consulting for GRAIL and Genentech. D.S. received research funding from Pfizer, Novartis, Syndax, Millenium Pharmaceuticals, Aileron Therapeutics, Bayer, and Genentech, owned stock in Biomarin, Amgen, Seattle Genetics, and Pfizer, served on the Board of Directors for BioMarin, and performed consulting/advisory board work for Eli Lilly, Novartis, Bayer, and Pfizer. The remaining authors declare no competing interests.

## Additional information

Sara A. Hurvitz[1,17 ✉], Jennifer L. Caswell-Jin [2,3,17], Katherine L. McNamara [2,3,4], Jason J. Zoeller[5], Gregory R. Bean [6], Robert Dichmann[7], Alejandra Perez[8], Ravindranath Patel[9], Lee Zehngebot[10], Heather Allen[11], Linda Bosserman[12], Brian DiCarlo[1], April Kennedy[13], Armando Giuliano[14], Carmen Calfa[8], David Molthrop[10], Aruna Mani[15], Hsiao-Wang Chen[1], Judy Dering[1], Brad Adams[1], Eran Kotler[2,3,4], Michael F. Press[16], Joan S. Brugge [5], Christina Curtis [2,3,4,18 ✉] & Dennis J. Slamon[1,18]

[1]David Geffen School of Medicine, University of California Los Angeles, Los Angeles, CA, USA. [2]Department of Medicine, Division of Oncology, Stanford University School of Medicine, Stanford, CA, USA. [3]Stanford Cancer Institute, Stanford University School of Medicine, Stanford, CA, USA. [4]Department of Genetics, Stanford University School of Medicine, Stanford, California, USA. [5]Department of Cell Biology, Harvard Medical School, Boston, MA, USA. [6]Department of Pathology, Stanford University School of Medicine, Stanford, CA, USA. [7]Central Coast Medical Oncology, Santa Maria, CA, USA. [8]Department of Medicine, University of Miami Miller School of Medicine, Miami, FL, USA. [9]Comprehensive Blood & Cancer Center, Bakersfield, CA, USA. [10]Florida Cancer Specialists & Research Institute, Orlando, FL, USA. [11]Comprehensive Cancer Centers of Nevada, Las Vegas, NV, USA. [12]City of Hope Comprehensive Cancer Center, Duarte, CA, USA. [13]FCPP Hematology/Oncology, San Luis Obispo, CA, USA. [14]Cedars-Sinai Medical Center, Los Angeles, CA, USA. [15]Genentech, South San Francisco, CA, USA. [16]Department of Pathology, Keck School of Medicine, University of Southern California, Los Angeles, CA, USA. [17]These authors contributed equally: Sara A. Hurvitz, Jennifer L. Caswell-Jin. [18]These authors jointly supervised this work: Christina Curtis, Dennis J. Slamon. ✉email: SHurvitz@mednet.ucla.edu; cncurtis@stanford.edu

