## [Peer Review File · Nature Communications]

This manuscript has been previously reviewed at another journal that is not operating a transparent peer review scheme. This document only contains reviewer comments and rebuttal letters for versions considered at *Nature Communications*.

Reviewers' Comments:

Reviewer #1:

Remarks to the Author:

the authors have addressed my comments

Reviewer #2:

Remarks to the Author:

The authors addressed my comments adequately. The correlative work was not based on pre-specified hypothesis and the findings will need further validation in future studies.

Specific comments:

1. The parameter estimate of beta from logistic models was presented for continuous variables. To make the magnitude of association easier to interpret, alternative odds ratio could be provided for every unit or every 10 unit increase of the continuous marker.

2. The enrollment number deviated from protocol. The plan was to randomized 120 to the three arms (40 per arm), but the final accrual was a few patients short in each arm. It helps to provide an explanation to ensure the quality of the clinical trial, even though the primary analysis was comparing pCR in the combination arm to historical control.

Reviewer #3:

Remarks to the Author:

I first want to acknowledge the efforts of the authors in the analyses originally presented in the manuscript as well as in the answers to the reviewers' questions. However, my main concern is still about the validity/utility of the results based on the expression analyses of the longitudinal samples presented here. I do not fully agree with some of the authors' statements in the review document, in which it is said that (sic) "we approach the problem with greater rigor and clearer understanding of its challenges than ever before" or "[the approach] is important to the overall direction of the field analysing these type of data". In my opinion, the limitations of interpreting quantitative data from bulk longitudinal samples with varying cells' content (specially when those changes are linked to the event of interest, i.e. response to therapy) is (or should be) very well known --and thus this cannot be considered new findings/information of a research paper. Similarly, the authors indeed describe/discuss these caveats e.g. when comparing expression results with those obtained with more accurate profiling techniques, but this study does not develop/apply a method to address these, so no clear 'direction' is given about how to solve those limitations. Of note, as the problem of not having a better experimental design for this type of data is indeed known, a number of bioinformatics methods have been previously described for (at least partially) addressing the issue from the processing side. Actually, some of them (see e.g. the approach to adjust per-gene expression bulk data for CD45 content in PMID:27386949) may be similar to that that the authors use to calculate the so-called "predicted ERBB2 and ESR1 signatures" -- although I am unsure of that as I did not find the details of the latter in the manuscript. Anyway, and specially given the current sample size, I do not believe that any analytical approach can significantly improve the interpretation of the data --which at the end of the day relies as a limitation of the study. As a general note, the authors may want to consider discussing these limitations as a premise of the study rather than a finding, and focus the results section on those data that they consider may be less affected (and thus reduce the

attention/extension given to the rest). I have no further questions about the issues raised in the first review, nor about the statistical/ clinical interpretation of the data.

Response to Reviewers

Reviewer #1 (Remarks to the Author):

the authors have addressed my comments

Reviewer #2 (Remarks to the Author):

The authors addressed my comments adequately. The correlative work was not based on pre-specified hypothesis and the findings will need further validation in future studies.

Specific comments:

1. The parameter estimate of beta from logistic models was presented for continuous variables. To make the magnitude of association easier to interpret, alternative odds ratio could be provided for every unit or every 10 unit increase of the continuous marker.

We agree, and have changed these estimates in Supplementary Tables 2 and 3 to be odds ratios of pathologic complete response for every unit increase of the continuous markers as recommended.

2. The enrollment number deviated from protocol. The plan was to randomized 120 to the three arms (40 per arm), but the final accrual was a few patients short in each arm. It helps to provide an explanation to ensure the quality of the clinical trial, even though the primary analysis was comparing pCR in the combination arm to historical control.

This was an investigator-sponsored study with funding provided by two pharmaceutical companies. Due to slower than expected enrollment, sponsors elected to end support for the study after four years when we had enrolled 130 of the 140 planned patients. Thus, enrollment in the two investigational arms were slightly lower than planned (36/40 in the TCL arm, 58/60 in the TCHL arm) and was 6 patients lower than planned in the control arm (TCH 34/40). Based on this reviewer comment, we now note in the second sentence of the Results that enrollment was 10 participants short of planned.

Reviewer #3 (Remarks to the Author):

I first want to acknowledge the efforts of the authors in the analyses originally presented in the manuscript as well as in the answers to the reviewers' questions. However, my main concern is still about the validity/utility of the results based on the expression analyses of the longitudinal samples presented here. I do not fully agree with some of the authors' statements in the review document, in which it is said that (sic) "we approach the problem with greater rigor and clearer understanding of its challenges than ever before" or "[the approach] is important to the overall direction of the field analysing these type of data". In my opinion, the limitations of interpreting quantitative data from bulk longitudinal samples with varying cells' content (specially when those changes are linked to the event of interest, i.e. response to therapy) is (or should be) very well known --and thus this cannot be considered new findings/information of a research paper.

Similarly, the authors indeed describe/discuss these caveats e.g. when comparing expression results with those obtained with more accurate profiling techniques, but this study does not develop/apply a method to address these, so no clear 'direction' is given about how to solve those limitations. Of note, as the problem of not having a better experimental design for this type of data is indeed known, a number of bioinformatics methods have been previously described for (at least partially) addressing the issue from the processing side. Actually, some of them (see e.g. the approach to adjust per-gene expression bulk data for CD45 content in PMID:27386949) may be similar to that that the authors use to calculate the so-called "predicted ERBB2 and ESR1 signatures" -- although I am unsure of that as I did not find the details of the latter in the manuscript. Anyway, and specially given the current sample size, I do not believe that any analytical approach can significantly improve the interpretation of the data --which at the end of the day relies as a limitation of the study. As a general note, the authors may want to consider discussing these limitations as a premise of the study rather than a finding, and focus the results section on those data that they consider may be less affected (and thus reduce the attention/extension given to the rest). I have no further questions about the issues raised in the first review, nor about the statistical/ clinical interpretation of the data.

We appreciate these comments on the limitations of our study, and agree with the reviewer that further bioinformatic work to attempt to disentangle bulk expression data with varying cellular contents is unlikely to add value. These challenges in interpretation are an intrinsic property of bulk longitudinal data, and we ultimately conclude that other technologies (e.g. in situ approaches) are better suited to address specific questions of change in tumor signaling and microenvironment over time, though we can draw some important conclusions (e.g. changes in immune subpopulations and stromal content) as outlined in the manuscript. Our hope is that our meticulous and frank discussion of these limitations will prove useful to other cancer biologists and oncologists designing longitudinal data collections (e.g. in the setting of clinical trials) who may wish to use bulk expression data to understand tumor evolution across treatment or metastasis; our impression has been that, however well-known these caveats should be, we have seen them glossed over or misunderstood. We are grateful for the reviewer's clear-eyed assessment of these issues as well.